# Analysis of Design and Fabrication Parameters for Lensed Optical Fibers as Pertinent Probes for Sensing and Imaging

**DOI:** 10.3390/s18124150

**Published:** 2018-11-27

**Authors:** Soongho Park, Sunghwan Rim, Ju Wan Kim, Jinho Park, Ik-Bu Sohn, Byeong Ha Lee

**Affiliations:** 1School of Electrical Engineering and Computer Science, Gwangju Institute of Science and Technology, 123 Cheomdangwagi-ro, Buk-gu, Gwangju 61005, Korea; shpark88@gist.ac.kr (S.P); rim@gist.ac.kr (S.R); 2Department of Biomedical Science and Engineering, Gwangju Institute of Science and Technology, 123 Cheomdangwagi-ro, Buk-gu, Gwangju 61005, Korea; scienc2@gist.ac.kr; 3Graduate School of Advanced Imaging Science, Multimedia and Film Chung-Ang University, 84 Heukseok-ro, Dongjak-gu, Seoul 06974, Korea; dkskzmffps@cau.ac.kr; 4Advanced Photonics Research Institute, Gwangju Institute of Science and Technology, 123 Cheomdangwagi-ro, Buk-gu, Gwangju 61005, Korea; ibson@gist.ac.kr

**Keywords:** fiber design and fabrication, optical fiber probe, optical fiber interferometer, optical fiber sensing and imaging, optical fiber coupling

## Abstract

A method for adjusting the working distance and spot size of a fiber probe while suppressing or enhancing the back-coupling to the lead-in fiber is presented. As the optical fiber probe, a lensed optical fiber (LOF) was made by splicing a short piece of coreless silica fiber (CSF) on a single-mode fiber and forming a lens at the end of the CSF. By controlling the length of the CSF and the radius of lens curvature, the optical properties of the LOF were adjusted. The evolution of the beam in the LOF was analyzed by using the Gaussian ABCD matrix method. To confirm the idea experimentally, 17 LOF samples were fabricated and analyzed theoretically and also experimentally. The results show that it is feasible in designing the LOF to be more suitable for specific or dedicated applications. Applications in physical sensing and biomedical imaging fields are expected.

## 1. Introduction

Optical fiber is a cost-effective flexible medium to deliver light with low loss. In addition to light transmission, many researchers have investigated its applications in various fields ranging from physical sensing to biomedicine [1,2,3,4,5,6,7]. One of the common applications is a lensed optical fiber (LOF) as a compact fiber probe. The LOF is a piece of optical fiber with an integrated lens at one end, so that it can collimate or focus the beam coming through a lead-in fiber. The fabrication process of such a fiber device has been established [8,9,10,11]. In many cases, the lens is fabricated by melting the end face of a fiber with a fiber fusion splicer. In general, a short piece of coreless silica fiber (CSF) is fusion-spliced prior to the lens fabrication. At the CSF region, the beam coming from the lead-in fiber is expanded before being collimated or focused by the lens. Owing to the small form factor, the LOF has been used in various fields, such as laser-fiber coupling, fiber pigtailed transceivers, and the optical interferometry system as a compact and efficient probe [2,6,7]. However, to the best of our knowledge, no explicit study has been given to the back-coupling efficiency while considering the working distance and beam diameter of a LOF [1,2,3,4,5,6,7,8,9,10,11,12,13]. Light reflected from the lens surface and back-coupling the lead-in fiber can seriously affect the optical system in some cases. In this paper, we focus on the utilization of a LOF as an optical element for an optical interferometry system and introduce the method or tool that can easily analyze the properties of the fiber-based optical element.

Largely, interferometry systems can be categorized into two types: the “Michelson-type” and the “Common-path-type”. In the case of a “Michelson-type” system, such as optical coherence tomography (OCT), a separated reference arm is used and the LOF acts only as a focuser for the sample arm probe [2]. In this case, the beam Fresnel-reflected at the lens surface of the LOF can interfere with the reference beam, which gives a noise or a ghost image. Therefore, we need to find a way that can minimize the amount of the beam back-coupled to the lead-in fiber after being Fresnel-reflected at the lens surface. However, in a “common-path-type” interferometer, there is no separated reference arm. Thus, the LOF has the role of both the reflector of the reference arm and the focuser of the sample arm [3,4]. In this case, there should be an appreciable amount of back-coupled beam to make interference with the sample beam. In other words, the same beam, back-coupled from the LOF surface, can act as an obstacle to be overcome, or as an indispensable element, depending on the application. Therefore, it is preferable to be able to simultaneously adjust all or parts of the LOF specifications, including the back-coupling from the LOF surface, in addition to the working distance and the beam diameter of the LOF focuser.

The major parameters affecting the back-coupling efficiency of a LOF are the length of the beam expansion region and the radius of the lens curvature. However, these parameters also influence the working distance and the beam diameter of the LOF probe. Therefore, we need to be careful in adjusting the back-coupling efficiency of the probe. Reversely, it can be thought that we can adjust the back-coupling efficiency and/or the beam diameter while keeping the same working distance. 

In this study, we investigate the optical properties of the LOF theoretically and experimentally, in terms of the length of the beam expansion region and the radius of curvature of the fiber lens. Firstly, the Gaussian ABCD method is presented as a tool for analyzing the evolution of the beam in a fiber and fiber devices. Due to the small diameter of the beam, the conventional ABCD ray-tracing method cannot give an accurate enough result, and the wave optics approach is much too heavy for handling. However, since the fundamental mode of a single-mode fiber (SMF) has a Gaussian-like intensity profile, we can use the ABCD method modified for the Gaussian beam with quite high accuracy, but without suffering from heavy computing time. Based on this Gaussian ABCD method, the fabrication parameters are simulated so that the probe has a different back-coupling efficiency while maintaining the same working distance. Finally, the working distances and the back-coupling efficiencies of 17 LOFs, fabricated to verify the proposed concept, are measured and evaluated. In addition, the variation of the beam diameter is examined with regard to the change of the back-coupling efficiency.

## 2. Lensed Optical Fiber (LOF)

The fundamental structure of a LOF for an optical probe is shown in Figure 1. The input beam from SMF is expanded before being focused by the lens, in order to provide an effective light-gathering power to the fiber lens. As the core of SMF is very small (<10 μm), without being expanded, the beam cannot be focused at a reasonable working distance in general. In order to fabricate the beam expansion region, a short piece of CSF is generally spliced in between, where the beam is expanded according to the numerical aperture (NA) of the SMF. The working distance (*L_f_*) and the beam waist (*w*_01_) are determined by the length of the beam expansion region (*L_c_*) and the radius of lens curvature (*R*). However, at the lens surface, a part of the beam is reflected and could be back-coupled to the lead-in SMF. For this case, the diameter and the propagation angle of the reflected beam at the SMF is important for calculating the back-coupling efficiency. 

### 2.1. LOF for Michelson-Type Interferometry

A Michelson-type optical interferometer system has a separated reference arm. Therefore, it is necessary to minimize the back-coupling from the lens surface of the LOF probe. Anti-reflection coating on the lens surface might be the best solution, since it removes the Fresnel reflection itself. However, from a practical point of view, coating on the fiber lens with a long piece of lead-in fiber is neither easy nor cost-effective. Therefore, as the second-best approach with the inevitable Fresnel reflection, we can think of minimizing the mode-coupling to the lead-in SMF by adjusting the structure parameters of the LOF. Figure 2 shows three types of typical LOFs. The LOF with a sufficiently large *R* (almost flat surface) is good, owing to the large beam size at the SMF. However, as shown in Figure 2a, its performance as a focuser is low due to the low lens power. Figure 2c shows the opposite case, that with a small *R* and a short *L_c_* we can have a good focuser. Moreover, in this case, the size of the beam reflected at the SMF is rather small, so that it gives fairly high back-coupling to the SMF. We can say the LOF optimized for a Michelson-type probe is the one in Figure 2b. The probe has a long *L_c_* and a large *R*, so that it can act as a good focuser but has a low back-coupling efficiency.

### 2.2. LOF for Common-Path-Type Interferometry

In a common-path-type system, the surface of the LOF lens acts as a reference reflector also. Therefore, in order to have an interferogram with good visibility, the design parameters should be adjusted in the direction of increasing the intensity of the beam back-coupled from the lens surface as much as possible. However, merely decreasing *R* from the probe of Figure 2b, designed as the best for the Michelson-type, can increase the back-coupling, but it affects the working distance *L_f_* heavily. Therefore, the CSF length (*L_c_*) needs to be adjusted simultaneously. As a result, we can say that the structure of Figure 2c, with a short *L_c_* and small *R*, is the best for the common-path-type system. It is noted that the adjustment of *R* and *L_c_* also affects the spot size of the probe beam, *w*_01_ in Figure 1.

## 3. Gaussian ABCD Matrix Analysis for LOF

When a beam passes through several optical elements in succession, the characteristics of the beam are changed. For a systematic approach, the ABCD matrix method was developed as a ray-tracing method in a paraxial regime of bulk optics [14]. In fiber optics, however, the conventional ABCD matrix method is not proper, due to the small size of the beam in the fiber and fiber devices. The small spot size asks for the consideration of wave diffraction. In fact, we can use wave optics or numerical calculation, but it might prove too much of a burden in handling. Fortunately, the intensity profile of the beam propagating through the core of an SMF is similar to a Gaussian shape [15], thus the beam coming out of a SMF and propagating in fiber devices can be approximated as a Gaussian beam.

Since the evolution of the wavefront of a Gaussian beam is equal to that of a spherical beam in the paraxial regime, the ABCD ray matrix of geometrical optics can be used to analyze the Gaussian beam in fiber optics [15,16,17]. The difference is that for a Gaussian beam, to treat the diffraction, the complex parameter is used for describing the beam. In a free space, the radius of wavefront curvature *r*(*z*) and the beam radius *w*(*z*) of a Gaussian beam can be expressed with a single parameter *q*(*z*), called the complex radius of curvature, as [17]
(1)$$\frac{1}{q\left(z\right)}=\frac{1}{r\left(z\right)}-i\frac{\lambda }{\pi w^{2}\left(z\right)n},$$
where λ is the free-space wavelength of the beam, and *n* is the refractive index of the medium. Further, when a Gaussian beam propagates along the optical axis *z* from position *z*_1_ to *z*_2_, it is known that the evolution of the parameter *q*(z) is effectively calculated by the matrix elements of the corresponding conventional ABCD transfer matrix, as [17]
(2)$$q_{2}=\frac{Aq_{1}+B}{Cq_{1}+D}.$$

In summary, by using the conventional ABCD transfer matrix, we can describe the evolution of a paraxial Gaussian beam, which is referred to as the Gaussian ABCD method.

### 3.1. Gaussian Beam Outside of LOF

The evolution of a Gaussian beam in each section of the LOF in Figure 1 can be represented with the transfer matrix *M_ij_* (subscripts *i* and *j* represent the propagation from the *i-*th to *j-*th plane). First, the beam propagation in the CSF region, from the SMF end to the lens surface, is described as a matrix *M*_12_ in Equation (3); the beam size increases, while its wavefront varies from a flat, to a rather spherical form. At the lens, only the wavefront curvature is changed without varying the beam size, which provides the lens power and can be described as *M*_23_. Similarly, *M*_34_ represents another free-space propagation to the focal plane located at *L_f_*. Finally, by multiplying everything, we have the total transfer matrix *M_out_*, as:(3)$$M_{out}=M_{34}M_{23}M_{12}=\left[\begin{array}{cc} A & B\\ C & D \end{array}\right]$$
where: $M_{12}=\left[\begin{array}{cc} 1 & L_{c}\\ 0 & 1 \end{array}\right]$, $M_{23}=\left[\begin{array}{cc} 1 & 0\\ \frac{n_{0}-n_{1}}{R\cdot n_{0}} & \frac{n_{1}}{n_{0}} \end{array}\right]$, $M_{34}=\left[\begin{array}{cc} 1 & L_{f}\\ 0 & 1 \end{array}\right]$.

The condition of focusing is that the radius of the wavefront curvature *r*(*z*) becomes infinite at the focal plane. Therefore, the working distance *L_f_* of the LOF can be calculated by finding the location *z* of the plane at which the real part of Equation (1) vanishes, which gives (see Appendix A):(4)$$AC+a^{2}BD=0,$$
with a constant coefficient of $a\equiv \frac{\lambda }{\pi w_{0}{}^{2}n_{1}}$,
where *w*_0_ denotes half of the modal field diameter (MFD) of the beam at the SMF. In general, the MFD is slightly larger than the physical core diameter of the SMF. Further, by using the imaginary part of Equation (1) and the evolution of *q*(*z*) given by Equation (2), the spot size (called the Gaussian beam waist) *w*_01_ at the *L_f_* of the LOF is calculated as (see Appendix A):(5)$$w_{01}=w_{0}\sqrt{\frac{n_{1}}{n_{0}}\frac{A^{2}+a^{2}B^{2}}{AD-BC}}$$

### 3.2. Gaussian Beam Back-Reflected from the Lens Surface

Similarly, the beam reflected at the lens surface of the LOF and then back-coupled to the SMF can be calculated with the Gaussian ABCD matrix method also. Since the inner surface of the lens acts as a concave partial mirror, the reflected beam is focused at point *L_b_*, indicated with the 5th plane in Figure 1, and then diverges again. The location of the focal plane *L_b_* can simply be calculated by using Equation (4), with the transfer matrix of:(6)$$M_{BC}=M_{25}M_{2}M_{12}=\left[\begin{array}{cc} A\prime & B\prime \\ C\prime & D\prime \end{array}\right],$$
where, $M_{12}=\left[\begin{array}{cc} 1 & L_{c}\\ 0 & 1 \end{array}\right]$, $M_{2}=\left[\begin{array}{cc} 1 & 0\\ -\frac{2}{R} & 1 \end{array}\right]$, $M_{25}=\left[\begin{array}{cc} 1 & L_{b}\\ 0 & 1 \end{array}\right]$.

At first, the beam propagates from the SMF to the lens with *M*_12_, and then it is reflected by the concave mirror with *M*_2_. Finally, there is another free propagation with *M*_25_. For simplicity, the lens surface of the LOF was considered as a spherical mirror. The Fresnel reflection at the lens surface was treated as uniform across the whole interesting lens area. Because of the low numerical aperture (NA) of the SMF, the incident angle to the lens is relatively small; thus we can assume the almost constant reflection coefficient.

The beam waist *w*_02_ at the 5th plane is simply calculated with Equation (5), and the matrix elements given by Equation (6). If the focal plane is located in front of the input plane, the beam is expanded again before launching to the lead-in SMF. Therefore, we need to make another free propagation, from the 5th to 1st plane, which gives the radius of the beam reflected at the lens surface and finally hitting the end surface of the SMF, as: (7)$$w_{2}=w_{02}\sqrt{1+{\left(\frac{\left(L_{c}-L_{b}\right)\lambda }{\pi w_{02}^{2}}\right)}^{2}}$$

In general, within a paraxial regime, the mode-coupling efficiency is obtained by calculating the overlap integral between the field distributions of two interesting modes [15]. In our case, since the core mode is approximated as having a Gaussian shape, the overlap integral is made with two Gaussian beams having different spot sizes, as well as different radii of wavefront curvatures. 

## 4. Experimental Methods

### 4.1. LOF Fabrication

As mentioned previously, it is necessary to adjust the back-coupling efficiency, the working distance, and/or the spot size of the beam at the working distance of a LOF probe, depending on specific applications. These adjustments can be made by two parameters: the length of the beam expansion region (*L_c_*), and the radius of curvature (*R*) of the fiber lens. In order to clearly see the effect of these two parameters, a total of 17 LOFs were fabricated with conventional SMF (Corning, New York, NY, USA, SMF-28) by an arc discharge of an optical fiber fusion splicer (FITEL, Tokyo, Japan, S183 PM) [9]. According to the fiber’s datasheet, the physical core diameter was 8.2 μm, and the MFD was 10.4 μm at a wavelength of 1550 nm. The refractive index of the CSF (Thorlabs, Newton, NJ, USA, FG125LA) was 1.444 at the same wavelength. Based on the simulation, the fabrication was intended to provide the same working distance, but different back-coupling efficiencies. 

### 4.2. Working Distance and Beam Diameter Measurements

In order to evaluate the working distance and the spot size, the knife-edge method was employed [18,19]. A sharp-edge mirror (Thorlabs, PFD10-03-M01) was placed perpendicular to the LOF, and the power of the beam coupled back to the fiber was measured. 

Figure 3a shows that the back-coupling becomes largest at the mirror position of 380 μm, which implies that the working distance of the LOF probe is 380 μm. Then, by laterally moving the sharp-edge mirror at the working distance, the spot size was measured. Figure 3b shows that the intensity of the back-coupled beam decreases monotonously; it was well-fitted with a sigmoid function. By taking a derivative of the fitted curve, the diameter of the beam focused at the working distance was obtained as the distance between the 1/e^2^ points. The blue curve in Figure 3b shows that the beam diameter is 21 μm.

### 4.3. Radius of Curvature Measurement of the Lens Surface

In order to calculate the working distance of a LOF probe, the geometric curvature of the lens was measured. In general, the surface shape of the LOF tip (fiber lens) is regarded as a part of a sphere. However, in practice, it can deviate from a sphere; its radius of curvature varies across the surface, where usually it depends on the distance from the optical axis. Therefore, it is necessary to specify the curve-fitting area of the lens first. In previous reports, the fitting areas were not meticulously considered [2,20]; therefore, a considerable error might occur in the measurement of *R*.

With the physical characteristics (numerical aperture and refractive index) of SMF, the interesting area of the lens surface was specified first. As shown with the red lines in Figure 4a, we know that the beam from the SMF propagates the CSF region with a small diverging angle and illuminates only a small portion of the lens surface, not the whole area. Then, by taking a binary image, the surface boundary was clarified (Figure 4b). Using the edge detection of the Laplacian filter, three points on the boundary were selected, as far apart from each other as possible (the red dots in Figure 4c). A circle passing through the three points was fitted. Finally, ten points (the blue dots in Figure 4c) were randomly selected along the boundary and used to evaluate errors in the curve-fitting (Figure 4d).

### 4.4. Beam Expansion Region Control and Measurement

The length of *L_c_* was controlled by using a home-made optical fiber cleaving system. A photograph of the system is shown in Figure 5. At first, a piece of SMF was held with a detachable optical fiber holder. Then, one side of the SMF piece was cleaved using a movable optical fiber cleaver, which could be shifted along the direction of the fiber length with a precision micrometer (Mitutoyo, Kawasaki, Japan, No. 350-251). Secondly, the cleaved SMF piece was moved, together with the detachable holder, to a fusion splicer and spliced with a piece of CSF. Finally, the other end of the CSF piece was cleaved after moving it to the original position of the system, together with the detachable holder. In this step, the cleaver was shifted by a desired length of *L_c_* with the micrometer. With this, the length *L_c_* could be controlled with micrometer accuracy. Similarly, with the same fabrication condition, forming the *R* of LOF could be kept within 3 µm. After fabricating a fiber lens, the final length of *L_c_* was measured using a charge-coupled-device (CCD) camera mounted on the splicer. The interface between the SMF and the CSF was visually distinguished by the refractive index difference between the two fibers. 

### 4.5. Back-Coupling Efficiency Measurements

The intensity measurement of the beam, reflected at the lens surface of the LOF and then mode-coupled back to the lead-in SMF, was performed with the setup in Figure 6. A 1550 nm CW laser was used as the light source; its intensity fluctuation was monitored using a 99:1 optical fiber coupler and an optical power detector 1 (Newport, Irvine, CA, USA, 818-IS-1). The back-coupled beam was measured with detector 2 (EXFO, Quebec City, QC, Canada, FPM-600). The fabricated LOF samples were connected to the measurement system one-by-one by using an optical fiber mating sleeve for the back-coupling efficiency measurements.

## 5. Results and Discussion

### 5.1. Radius of Curvature of the LOF

As was presented in Figure 4, a microscopy image of a LOF was captured with the CCD camera of the fusion splicer. The discrepancy between the fitted circle and the 10 points taken along the boundary of the LOF image in Figure 4b was evaluated. The root mean square (RMS) of the discrepancy of the whole fabricated probes was smaller than 200 nm, which implies that the curve-fitting was quite reasonable. The CCD had only 720 × 480 image pixels. If a high-resolution image was available, the accuracy of the measurement could have been improved further. 

### 5.2. Working Distance and Beam Diameter Simulations

The working distance was numerically calculated using the Gaussian ABCD matrix as a function of *L_c_* and *R* presented by the color code in Figure 7a. In the figure, the white contour represents the design parameters giving the same working distance of 380 μm, which implies that the same working distance can be obtained with various combinations of *L_c_* and *R*. It is worth noting that the straight white line in the lower-right part represents a working distance of 0 μm. Below the white line, any combinations cannot provide a focusing probe. The beam diameter or spot size of each LOF was also calculated at the working distance. Figure 7b shows that the spot size of the focused beam can be appreciably controlled without changing the working distance by taking various combinations of *L_c_* and *R*.

It is also worth noting with the white contour of Figure 7a that for a given *L_c_*, there were two *R* values giving the same working distance. However, the *R* value at the right side of the contour led to a significantly larger beam diameter than the left side, as can be seen in Figure 7b. Therefore, the fabrication and experiment were performed with the conditions of the left side of the contour. The general trend was that with the increase of *L_c_*, a larger *R* was required to maintain the same working distance. In addition, at the same working distance, the increase of *L_c_* yielded a smaller spot size.

### 5.3. Working Distance and Beam Diameter Measurements

Based on the simulations presented in Figure 7, a total of 17 LOFs have been designed and fabricated to have the same working distances of 380 μm with different fabrication parameters. First, *L_c_* and *R* of each fabricated LOF were measured and depicted in Figure 7 with black dots. The working distance of each LOF was then calculated with the experimentally measured *L_c_* and *R*, and compared with the directly measured one. As listed in Table 1, the average working distance was calculated to be 381.47 μm, slightly larger than the targeted value of 380 μm. The standard deviation was as small as 21.95 μm. In the direct measurement, the average and standard deviation of the working distance were 368.06 μm and 27.30 μm, respectively.

Although there is some error between the design and the fabrication, the experimental result was rather well-matched with that of the calculation. The experimental results implied that the parameters of the LOF could be adjusted, while keeping the same working distance. As an example, in order to fabricate the LOFs with a working distance of 380 μm, the beam expansion length *L_c_* can be varied up to 86 μm, from 294 μm to 380 μm, and the radius of curvature *R* can be varied up to 8 μm, from 66 μm to 74 μm. In practice, it is challenging to maintain a constant shot–to–shot arc discharge power of the fusion splicer during the fabrication of fiber lenses. However, the fabrication can be more precise by using the latest fiber fusion splicer or other precision techniques, such as UV curing resin [11], micro-structuring with femtosecond 3D printing [21], etc.

The beam diameter at the working distance of each fabricated LOF was experimentally measured and compared with the calculated value in the right column of Table 1. As shown in the table, for the probes of similar *R*, a larger *L_c_* yielded a smaller beam diameter. It is worth noting that among the LOFs with similar working distances (samples 1 and 16 in Table 1), there was a variation in the beam size by approximately a factor of two, from 21.2 μm down to 11.9 μm. In other words, it can be said that a specific beam diameter can be achieved while maintaining the same working distance, by taking a specific combination of *L_c_* and *R*.

### 5.4. Back-Coupling Efficiency Measurements

The back-coupling efficiency of each fabricated LOF was experimentally measured and compared with the one calculated with the measured *L_c_* and *R*. Figure 8a shows that a higher back-coupling efficiency is obtained with a smaller *L_c_* and a larger *R*, which might be desirable for the implementation of a common-path-type fiber interferometer. The maximum back-coupling can be obtained by reducing *L_c_* down to zero and increasing *R* infinitely, which represents a cleaved SMF without the CSF piece and fiber lens. However, it does not have any working distance, hence it cannot be used as a probe. Considering the same working distance, we know that the smaller *L_c_* asks for a smaller *R* in the figure.

Conversely, the LOF with a smaller back-coupling efficiency, preferred for implementing a Michelson-type interferometer, can be obtained using a larger *L_c_* and a smaller *R*. However, considering a reasonable working distance, the larger *L_c_* asks for a larger *R*, as indicated with the white contour in Figure 8a. For the 17 LOF samples, the back-coupling efficiencies were measured and compared with the calculated ones in Figure 8b. The Pearson’s correlation coefficient was calculated to reveal the correlation between them, which yielded approximately 0.93. Even though there is some difference in the magnitudes, they are well-matched in the trend.

In the experiments, the coupling efficiency was measured as the ratio of the beam-intensity back-coupled to the SMF after reflection from the lens surface to the one reflected from the end surface of an SMF only cleaved without CSF. The difference in the coupling efficiency of Figure 8b could be attributed to the following factors. Firstly, the back-coupling efficiency was calculated by considering the overlap integral between two Gaussian mode profiles. In the calculation, the core mode of the SMF was approximated as a Gaussian beam, and the diameter of the beam was considered as the MFD of the core mode. Secondly, a considerable connection loss could exist in the system. It is thought that a more delicate consideration should be made to improve the measurement accuracy. However, with this research, we can say that the Gaussian ABCD method is proper enough to investigate the evolution of the beam in fiber devices and to evaluate the optical properties of a LOF probe.

## 6. Conclusions

We have presented the Gaussian ABCD method that is effective in analyzing the optical properties of the beam in a fiber and fiber devices. As a simple optical fiber probe, a LOF has been analyzed, including its design and fabrication parameters. With 17 LOFs, designed and fabricated to have the same working distances but with different back-coupling efficiencies, the fiber probe properties were evaluated and the validity of proposing the Gaussian ABCD method was confirmed. 

By splicing a short piece of CSF to a lead-in SMF and forming a lens at the far end of the CSF, the LOF probe could be easily fabricated. The working distance *L_f_* of the probe and the spot size *w* of the focused beam were simulated with the length *L_c_* of the CSF piece and the curvature *R* of the fiber lens. Based on the simulation, 17 LOFs were fabricated to have the same *L_f_* of 380 μm but different *L_c_* and *R*. With the simulation and experiment, we got several results. The same *L_f_* could be maintained while increasing the CSF length from 294 μm to 380 μm, only with a slight increase in *R*. However, the increase of *L_c_* decreased the *w*_01_ and weakened the back-coupling from the lens surface to the lead-in fiber. Similarly, the amount of back-coupling to the lead-in fiber could be increased or decreased by a large amount without changing the *L_f_*, but just taking a different combination of *L_c_* and *R* of the probe. 

By introducing the complex radius of curvature of the Gaussian beam into the conventional ABCD matrix, the evolution of the beam coming from a fiber and then propagating, reflecting, and/or being focused by a lens could be well-analyzed. The intensity distribution of the beam guided in a fiber and evolving through fiber devices could be approximated as a Gaussian shape. Owing to the small diameters of fiber devices, the paraxial approximation was utilized. It is believed that these results pave the way for strategic fabrication of a dedicated fiber probe for a specific application. For example, we can fabricate an optical fiber probe having a back-coupling efficiency more suitable for implementing a Michelson-type or a common-path-type optical interferometer. Various applications are expected in the fields of biomedical imaging and smart sensing.

## Figures and Tables

**Figure 1 sensors-18-04150-f001:**
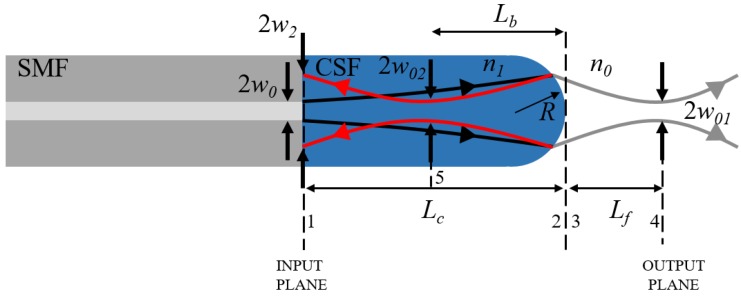
Schematic of a lensed optical fiber (LOF) structure. *L_c_*: length of coreless silica fiber (CSF), *L_f_*: working distance or focal length, *L_b_*: focal length of reflected beam, *R*: radius of lens curvature, *w*_0_: half of the mode-field diameter (MFD) of single-mode fiber (SMF), *w*: beam spot at each interesting position, *n*: refractive index of the medium.

**Figure 2 sensors-18-04150-f002:**
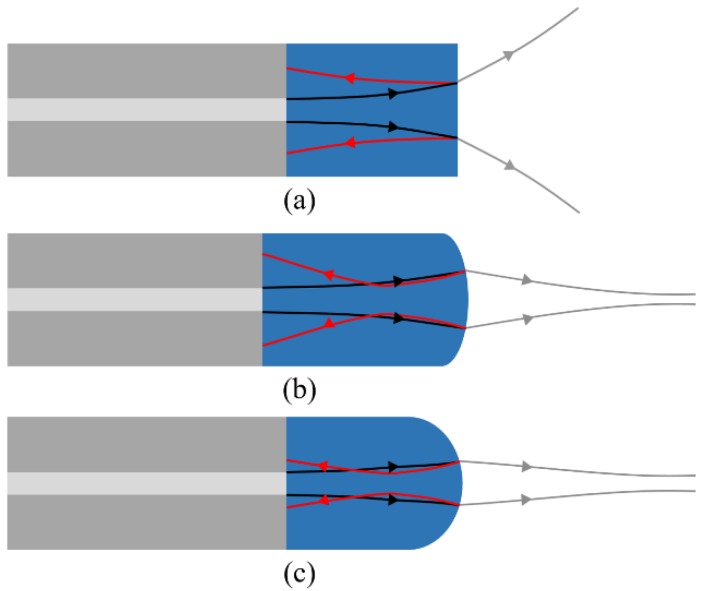
Various LOF structures. (**a**) LOF with a too-large radius of lens curvature (*R*). (**b**) LOF optimized as a probe for a Michelson-type interferometer. (**c**) LOF with a small *R* for a common-path-type interferometer. (**b**,**c**) are designed to have the same working distance but different back-coupling efficiency (black line: incident beam, red line: beam reflected at the LOF surface).

**Figure 3 sensors-18-04150-f003:**
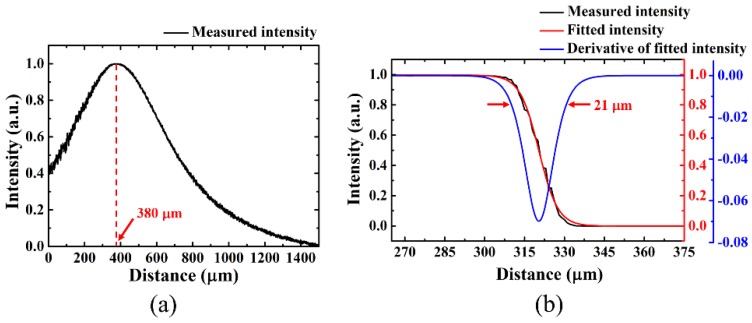
Performance evaluation of a fabricated LOF. (**a**) Back-coupled beam intensity measured as a function of the axial distance of a sharp-edge mirror. (**b**) Back-coupled beam intensity measured as a function of the lateral displacement of the sharp-edge mirror, located at the working distance of the LOF.

**Figure 4 sensors-18-04150-f004:**
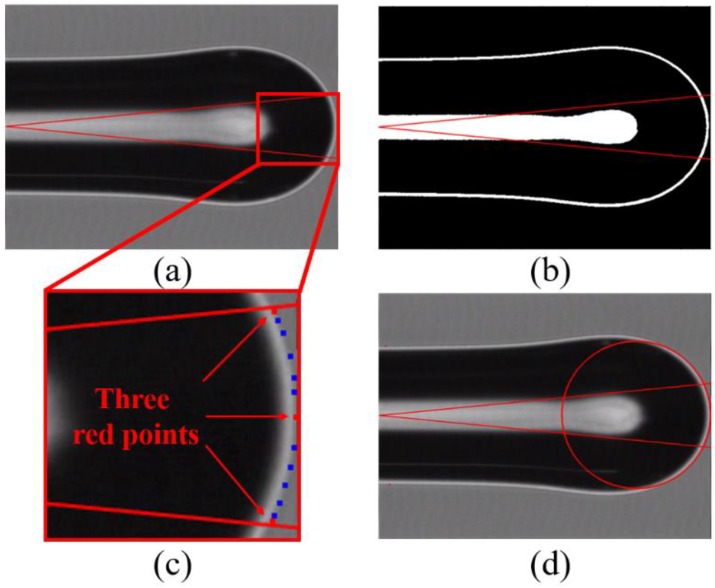
The radius of curvature measurement for a LOF lens surface. (**a**) Microscope image of a fabricated LOF, (**b**) binary image, (**c**) magnified image, and (**d**) curve-fitted image. The red lines represent the area where the beam actually propagates. The red dots in (**c**) were taken from the binary image of (**b**), and used to get the fitting circle of (**d**).

**Figure 5 sensors-18-04150-f005:**
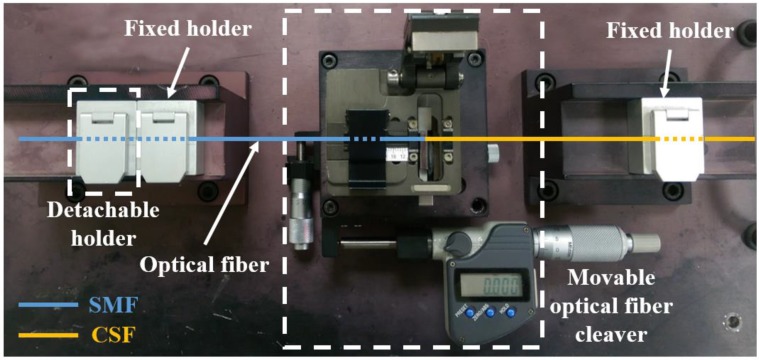
Photograph of the precise cleaving system used to control the length of the LOF’s beam expansion region (*L_c_*). The cleaver could be shifted laterally with a precision micrometer.

**Figure 6 sensors-18-04150-f006:**
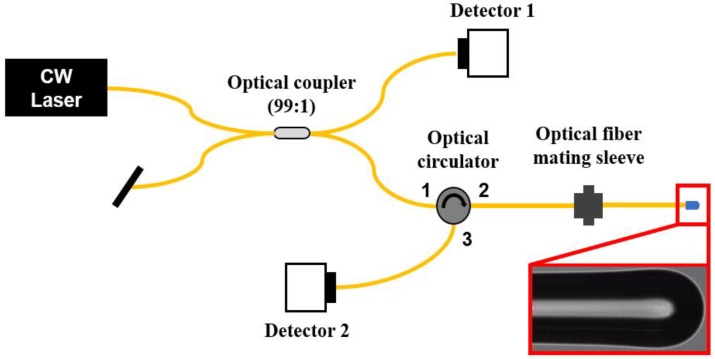
Schematic of the back-coupling efficiency measurement system. The intensity of the beam, reflected from the lens surface of a LOF and then back-coupled to the lead-in SMF, is measured.

**Figure 7 sensors-18-04150-f007:**
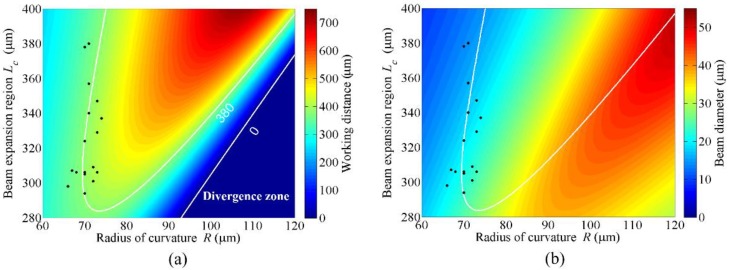
Working distances (**a**) and beam diameters (**b**) of LOF probes numerically calculated as a function of parameters *L_c_* and *R* by using the Gaussian ABCD matrix method. The black dots represent the measured parameters of fabricated 17 LOFs. The white contour represents the combination of parameters giving the same working distance of 380 μm.

**Figure 8 sensors-18-04150-f008:**
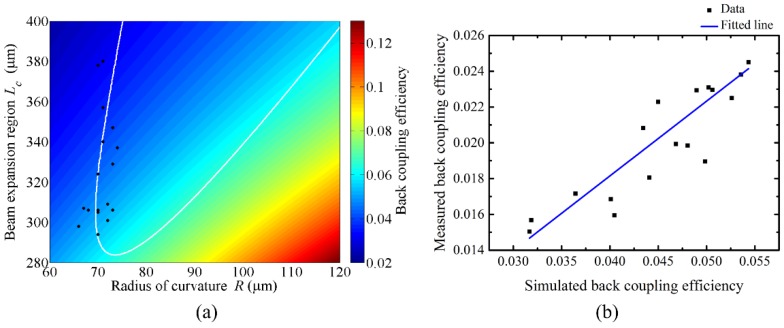
Back-coupling efficiency of a LOF. (**a**) Simulated and color-coded in terms of *L_c_* and *R*. The black dots represent the fabrication parameters measured with 17 fabricated LOFs. The white contour represents the various combinations for having a working distance of 380 μm. (**b**) Measured and compared with simulated ones. The blue solid line is a fitted line.

**Table 1 sensors-18-04150-t001:** Calculated and measured working distances and beam diameters of a total of 17 fabricated LOFs.

Sample No.	*L_c_* (μm)	*R* (μm)	Working Distance (μm)			Beam Diameter (μm)		
			Calculated	Measured	Error	Calculated	Measured	Error
1	294	70	381	350	31	22.9	21.2	1.7
2	298	66	350	384	34	19.4	21.2	1.8
3	301	72	399	389	10	23.5	19.5	4
4	305	70	384	380	4	21.4	21.2	0.2
5	306	68	367	379	12	19.8	18.7	1.1
6	306	70	384	381	3	21.3	17.8	3.5
7	306	73	409	386	23	23.6	22.0	1.6
8	307	67	357	359	2	19.0	17.8	1.2
9	309	72	402	375	27	22.4	17.8	4.6
10	324	70	380	338	42	19.0	15.0	4
11	329	73	408	439	31	20.4	22.1	1.7
12	337	74	415	381	34	20.1	15.4	4.7
13	340	71	382	351	31	17.8	15.0	2.8
14	347	73	399	361	38	18.3	22.0	3.7
15	357	71	370	330	40	16.1	11.0	5.1
16	378	70	345	347	2	13.9	11.9	2
17	380	71	353	327	26	14.2	16.1	1.9
Average			381.47	368.06	22.94	19.6	18.0	2.7
*σ_s_*			21.95	27.30	14.29	2.9	3.5	1.5

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
