# Peer review of "Analysis of Design and Fabrication Parameters for Lensed Optical Fibers as Pertinent Probes for Sensing and Imaging"

_sensors, 2018, doi:10.3390/s18124150_

Round 1
Reviewer 1 Report
The paper “Analysis of Design and Fabrication Parameters for Lensed Optical Fibers as Pertinent Probes for Sensing and Imaging” investigated the fabrication of lensed optical fiber (LOF), which includes a short piece of coreless silica fiber (CSF) and a lens formed at the end of CSF via fiber fusion splicer. The authors also simulated the relation of the length of CSF and radius of lens curvature, to the working distance and spot size of the focused light. Furthermore, experimental results measured through 17 samples well-matched the theoretical calculations. The paper is well-written, hence I recommend its publication on Sensors. Some questions are as below:
How is the repeatability of LOF fabrication, i.e., if the same fabrication condition is used, what are the errors for the working distance and spot size? And is the error acceptable for actual applications?
What is q1 and q2 in equation (2)?
Author Response
First of all, we would like to sincerely thank the reviewer for the efforts and time in reviewing our work. We highly appreciate the valuable comments and suggestions, which contributed to improve the quality of our work. The responses to the reviewer’s comments are made and submitted in a separate file. Please, check the attached file.

Reviewer 2 Report
the present paper should be improved in presentation and in result before to consider it
Author Response

(The authors gave the same response as above.)

Reviewer 3 Report
The work entitled “Analysis of Design and Fabrication Parameters for Lensed Optical Fibers as Pertinent Probes for Sensing and Imaging” investigates the potentiality of the ABCD matrix method in describing the optical propertied of a Lensed Optical Fiber (LOF). The simulation results are supported by experimental measurements.
The work is well-written, well-structured and present interesting results, which are clearly illustrated. However, some points can be improved.
Minor revisions:
1) The authors clearly explain how they proceed for the measurement of R. It would be interesting to describe how Lc is controlled and measured.
2) In table 1 the columns Lc and R represents experimental measurements. What is the associated experimental error. What are the expected uncertainties on the working distance and beam diameter by propagating the experimental error according to the abcd matrix method?
3) It would be interesting to know how well the authors can control the targeted dimension of Lc and R (possibly quantify).
4) With regards lines 315 to 317. If the measurement was carried out by changing an optical connection (e.g. replacing the LOF with a SMF 28) some difference in the magnitudes of figure 7b should clearly be expected. If this was the way in which the authors proceeded, it should be clearly stated.
Another possible procedure is to cleave the SMF28 pigtail of the LOF in order to make the comparison without touching any optical connection (by recording the back-scattered light before and after the cleaving). If this is what the authors have done it should be clearly stated. However, in this case, the discrepancies in the magnitudes of figure 7b cannot be explained by losses at optical connections anymore (with maybe the exception of the splice between SM28 and the CSF). In this case comment would be required.
Author Response

(The authors gave the same response as above.)

Reviewer 4 Report
In the submitted article, the authors present a method for adjusting the working distance and the spot size of a fiber by designing properly an integrated lensed tip in terms of length and curvature radius. In addition, the back-coupling is considered and treated using a simple but effective ABCD model, and a lot of samples have been fabricated and analyzed, comparing experimental data with simulations. The subject is well-presented and described, however, there are a few points to be addressed in order to make the paper accepted for publication:
- in equation (1), a square power is missing for the parameter w(z). The correct expression should contain w2 instead of w.
- It would be useful to derive the solution for Lf as a function of the beam and lens parameters, from eq. (4).
- Please check eq. (5), it seems not to be the correct solution.
- In table 1, please insert the experimental errors for the measured quantities.
- Please insert error bars to the experimental points in plots in figures 6 and 7.
- In figure 5 the mirror at the output of the fiber end is missing.
- In section 4.3 it is not clear how the aperture angle in Figure 4 was chosen in order to select the lens cross-section for the estimation of the average radius of curvature.
Author Response

(The authors gave the same response as above.)

Round 2
Reviewer 2 Report
in this second version the paper can be accepted
Reviewer 4 Report
The authors have properly addressed every point of the revision. The paper has been amended and improved, and we believe it can be accepted for publication as it is.